# **Evaluation of Peaks-Over-Threshold Method**

Soheil Saeed Far<sup>1</sup> and Ahmad Khairi Abd. Wahab<sup>1,2</sup>

<sup>1</sup>Faculty of Civil Engineering, Universiti Teknologi Malaysia (UTM), 81310, Skudai, Johor Bahru, Malaysia. <sup>2</sup>Center for Coastal & Ocean Engineering, Research Institute for Sustainable Environment, UTM, Jalan Sultan Yahya Petra, 54100, Kuala Lumpur, Malaysia.

*Correspondence to:* Ahmad Khairi Abd. Wahab (Email: akhairi@utm.my)

Abstract. Two extreme wave analysis models, namely Peaks-Over-Threshold (POT) and Generalized Pareto Distribution (GPD), were developed in order to improve the POT model and highlight merits and limitations of the two models. Studies have shown that the POT model was not equipped with a suitable approach to determine a true threshold value. This paper proposed an approach to specify the most suitable threshold value for the POT model, which is called Hybrid method. In

- 5 addition, until now the MIR (minimum ratio of residual correlation coefficient) criterion has been used as a goodness-of-fit method in the POT model. However, the examinations on the method represented that MIR is not always a stable approach in determining true distribution function. This paper proposed an alternative approach instead of the MIR criterion method, it is called Norm of Residuals, and its credibility was examined by the Chi-Square test. The results drawn from this study also demonstrated that the Hybrid method completely matched with the POT model, and the threshold obtained by this method is
- credible, moreover, the Norm of Residuals method is completely stable in determining the best fitting distribution for the POT 10 model.

Keywords: POT (Goda) model, Generalized Pareto Distribution (GPD) model, threshold value, Chi-Square test.

## 1 Introduction

Currently, many types of structures are designed for different purposes in coastal and offshore regions. The force of storm waves 15 usually influences the life span of these structures, compelling engineers to design structures to withstand these destructive waves. Therefore, the estimation of extreme waves is important in the design phase of structures in the marine areas. However, prior to the phase of designing, the probability of storm waves during the shelf life of structures should be projected. The estimation of a probable wave height is called return period of storms, and the employed method is generally known as extreme value analysis. Fundamentally, the aim of extreme value analysis is to specify the probabilities of exceedance and non-exceedance of wave heights, and non-exceedance probabilities are known as return periods.

There are several methods to estimate extreme values; two commonly used models, namely Peaks-Over-Threshold (POT) and Generalized Pareto Distribution (GPD) were developed in this study.

One crucial step in developing the POT model is to select the best fitting distribution function for dataset. To determine a best fitting distribution, two methods were introduced by Goda (1988); namely correlation coefficient method, and the MIR

<sup>20</sup> 

(minimum ratio of residual correlation coefficient) criterion method. However, examinations represented that the MIR criterion method has shown some instabilities in determining correct results when it comes to choosing a true distribution function. This study proposed an alternative approach instead of the MIR criterion method, it is called norm of residual (N.o.R).

Selecting an appropriate threshold value is important because the estimated extreme events by the POT model are sensitive to the changes in threshold value. Goda (2000) pointed out that in selecting a suitable threshold, users should refer to previous studies and engineering judgment. Li et al. (2012) followed the same procedure to determine a true threshold value. The authors provided a list of several extreme analysis studies, which had been conducted on the same data, and came to a conclusion with one of the thresholds used by one of those studies. However, in absence of a reliable threshold method, the other authors in the list of Li et al. (2012) could claim of the credibility of their threshold values. Moreover, if there is no preceding study for a

10 particular data, engineering judgment alone may not be sufficient when selecting a suitable threshold. At present, there is still no appropriate approach for selecting a threshold value for the POT model. A hybrid method consists of a two-part procedure; a mean residual life (MRL) plot, and a graph of several thresholds, was proposed in this paper to specify a suitable threshold value for the POT model.

Based on the definition by Grimshaw (1993), the GPD is a two-parameter family of distribution, which is applied to a data
model exceeding a known threshold value. The model was introduced for the first time by Pickands (1975) and is known to
be a stable distribution for exceedances beyond a threshold. The model has been presented by many authors; such as, Davison (1984), Smith (1985), Van Montfort & Witter (1985), Hosking & Wallis (1987), and Coles (2001).

The data that was applied in this study comprised a set of wave heights gathered from shipboard observations over a 41-year period from 1949 to 1989 in the South China Sea. Since 1989, the process of data acquisition was stopped, and the modern observation was replaced by satellites. In this study, only the observed data during the 41-year period has been used, due to the focus of this study on the merits and limitations of the two models, and improving the POT model. Moreover, there is a previous study on the same data, which could be referenced and compared with the new findings.

The data were collected within the Marsden Square Numbers 2554, 2555, 2564 and 2565 (the rectangular area within the magnified section of Fig. (1)) in the Federal Territory of Labuan off the coast of Sabah, Malaysia. The Labuan metropolitan
coast is located at the South China Sea on the continental shelf of Malaysia. National Climatic Data Center, USA compiled the dataset, which is known as Labuan data. The Labuan data are available from the wave database of the Drainage and Irrigation Department, Malaysia.

The Labuan data are a set of wave height records in which the time intervals between the successive data are considered to be at least three days to secure the required independence of the data. To fill the large gaps between the successive data; e.g.,

30 more than 4 days, the linear interpolation method was used since the data varied relatively slowly. A time interval of two to four days between successive peaks was recommended by Mathiesen et al. (1994) to maintain the condition of independency.

Figure 1. Location of Federal Territory of Labuan, and the surrounding offshore area.

The northeast monsoon waves (same direction with the smaller arrow in the magnified part of the map in (Fig. 1)) were the predominant waves among the wave heights for the four seasons in the region. About 65 percent of the waves during this season approached from between 30 to 60 degrees (Syed Abdullah, 1992). Therefore, to fulfill the requirement for homogeneity in the data, only storms in the compass direction of 30 to 60 degrees (northeast monsoon waves) were employed in the analysis.

5 The purpose of the foregoing tasks were to obtain the conditions of independent and identically distributed (iid) measurement for the Labuan data. The Labuan data was evaluated and used for the first time by Coastal and Offshore Engineering Institute, Universiti Teknologi Malaysia. (2009) as a report on extreme wave analysis for the Marine Department of Malaysia. The results of the report (Table 1) represent the estimated extreme wave heights for several return periods. The authors employed only one distribution function (Gumbel distribution) in their analysis without considering any threshold value.

Table 1. Extreme wave heights estimated by Gumbel distribution and the Labuan data (UTM Archive).

| Return Periods (yr)          | 2    | 5    | 10   | 20   | 30   | 50   | 100  |
|------------------------------|------|------|------|------|------|------|------|
| Significant Wave heights (m) | 3.08 | 3.57 | 3.89 | 4.20 | 4.38 | 4.60 | 4.90 |

<sup>10</sup> The following objectives were discussed in this paper: i) developing the proposed Hybrid method to specify true threshold value for the data for the POT model; ii) conducting the Norm of Residual method for the POT model to select the best fitting distribution function for the dataset; and iii) developing the modified POT (POT with the two proposed methods by this study) and GPD models, and comparing their results to evaluate the proposed methods and highlight merits and limitations of the two models.

This paper has been divided into four sections. Following this introduction, Section 2 provides summaries about the two models, and proposed methods for the POT model. Discussion about the return values and figures for both models, the goodnessof-fit tests, and the proposed methods are presented in Section 3. Finally, Section 4 contains the conclusions drawn from this study.

## 5 2 Methodology

#### 2.1 Generalized Pareto Distribution Model

Generalized Pareto Distribution model could be fitted for the observed wave height data with considering the following assumptions: The data are considered as exceedances from a specific threshold, which are a sequence of independent and identically distributed (iid) measurements, x<sub>(1)</sub>,...,x<sub>(k)</sub>. If exceeded data over a particular threshold u are defined as y<sub>(j)</sub> = x<sub>(j)</sub> - u for
10 j = 1,...,k, therefore, by definition, a random variable of y<sub>(j)</sub> may be regarded as independent realization whose distribution is approximated by one of the members of the Generalized Pareto family (Coles, 2001).

The model has been evaluated and used by many authors: Smith (1984) developed and presented the model in a form of comparison method with a statistical model, Van Montfort & Witter (1985) embedded exponential distribution in GPD to check the adequacy of the exponential distribution for a set of data, Hosking & Wallis (1987) studied about several estimators for the

- 15 GPD model, Moharram et al. (1993) estimated the river flood by the GPD model, Castillo & Hadi (1997) used the model for estimating wave height, Pandey et al. (2001) studied extreme wave velocity by the model, Coles et al. (2003) employed extreme analysis for estimating rainfall, Pandey et al. (2004) used the model to evaluate the sea level, Öztekin (2005) in estimating river discharge, Jagger & Elsner (2006) in studying hurricanes, and Moisello (2007) employed the model to estimate the extreme rainfall.
- 20 The generalized Pareto distribution function has the following formula,

$$F(x) = 1 - \left(1 - \frac{\xi x}{\sigma}\right)^{1/\xi} \tag{1}$$

In general, the two parameters of GPD are known as: the scale parameter  $\sigma$  ( $\sigma > 0$ ), and the shape parameter  $\xi$  ( $-\infty 

considered as  $n_{(u)}$  observations, exceeded over a threshold u, and  $x_{max}$  is the greatest value of  $x_i$ . Therefore, Eq. (2) is termed as the MRL formula, and denoted as,

$$\left\{ \left( u, \frac{1}{n_u} \sum_{i=1}^{n_u} (x_i - u) \right) : u < x_{max} \right\}$$

$$\tag{2}$$

Second Technique of Determining Threshold Value: The second method to determine threshold value was introduced as the complementary technique to evaluate the stability of parameter estimation (Coles, 2001). In general, the method of work is to fit GPD for a range of thresholds, and searching for the stability of parameter estimates. Therefore, if GPD is a reasonable model for data exceeding a threshold u, then, the model should be reasonable for higher threshold  $u_1$ . The argument suggests two graphs of shape and modified scale against different threshold values with confidence intervals for each of these quantities. The two graphs are called probability and quantile plots. The technique is carried out after the model estimation (Figs. 4,5).

## 10 2.1.2 Determining The Parameters

Once the threshold value has been specified, estimating the shape and scale parameters would be the next step of developing the model. Based on the size of the data, different parameters estimators methods can be applied to estimate the GPD parameters. Mackay et al. (2011) conducted a comparison of several estimators by Monte Carlo simulation, and argued that likelihood-moment (LM) estimators is the most substantial method with the lowest bias and variance. Zhang & Stephens (2009) proposed

15 an estimation procedure, and called it modified likelihood moment estimators (LME). Dupuis & Tsao (1998) proposed a hybrid estimators of two well-known estimators: method of moments (MOM), and probability weighted moments (PWM). The method was derived by incorporating a simple auxiliary limitation of the estimates.

In addition, de Zea Bermudez & Kotz (2010) carried out an extensive study on several types of methods for estimating the GPD parameters. The authors argued that the success of GPD on a set of data depends substantially on the process of parameter estimate, and maximum likelihood (ML) estimators has been the most popular method among other estimators. However, the ML estimators exists only for the shape parameter  $\xi \le 1$  because for  $\xi > 1$  the log-likelihood becomes infinity. In this study, the maximum likelihood (ML) estimators was employed to estimate the parameters. The log-likelihood formula for  $\xi \ne 0$  was derived from Eq. (1) as follows,

$$\ell(\sigma,\xi) = -n\log\sigma + \left(\frac{1}{\xi} - 1\right)\sum_{i=1}^{n}\log(1 - \frac{\xi x_i}{\sigma}) \tag{3}$$

25 As foregoing discussion, numerical techniques are required to estimate the shape and scale parameters.

#### 2.1.3 Return Level and Confidence Intervals

Subsequent to the estimation of the shape and scale parameters, the return levels should be estimated. Coles (2001) argued that the interpretation of extreme value models in terms of return levels are usually more convenient than using individual parameter

15

values. Therefore, we assume that GPD with defined shape and scale parameters is an appropriate model for the exceeded data over a threshold u to estimate m-observation return level,  $x_m$ ,

$$x_m = u + \frac{\sigma}{\xi} \left[ (m \cdot \zeta_u)^{\xi} - 1 \right] \tag{4}$$

where σ and ξ are the scale and shape parameters, respectively. And ζ<sub>u</sub> is described as ζ<sub>u</sub> = Pr{x<sub>i</sub> > u}, in which x<sub>i</sub> stands
for the wave heights, and u is the threshold value. Consequently, the return level x<sub>m</sub>, which is exceeded on average once every m observations being the solution of Eq. (4). Also, the parameters m and ζ<sub>u</sub> can be evaluated by Eqs. (5, 6) as follows,

$$m = ARI \frac{N_t}{k} \tag{5}$$

in which ARI and  $N_t$  represent, respectively, the Average Recurrence Interval (return period), and the total number of data during a period of k years. Equation (6) can be employed to compute  $\zeta_u$ ,

$$10 \quad \zeta_u = \frac{N}{N_t} \tag{6}$$

By substituting Eqs. (5, 6) into Eq. (4),  $x_m$  can be computed as,

$$x_m = u + \frac{\sigma}{\xi} \left[ (ARI. \frac{N}{k})^{\xi} - 1 \right] \quad for \ (\xi \neq 0)$$

$$\tag{7}$$

where N is the number of events exceeding a threshold u. Subsequently, the confidence interval for the return graph should be computed. There are several methods to derive confidence intervals for  $x_m$ , variance-covariance matrix, and delta methods are reliable methods (Davison & Smith, 1990; Coles, 2001; Pickands, 1975).

To check the quality of the modeling, probability plot, return level plot and quantile plot are usually evaluated (Coles, 2001).

#### 2.2 Peaks-Over-Threshold Model (Goda Method)

The Peaks-Over-Threshold (POT) model was introduced by Goda (1988), and has been developed in this study for further evaluation. The model can be developed with a long length dataset to obtain the results in a smaller range of confidence
intervals (Goda, 2010). Following is a summary of developing the model which were described by Goda (1988, 2000, 2010), as well as the proposed methods added by this study in order to improving the model:

- 1. After arranging the data in descending order, specifying the threshold value is the next step of the analysis.
- 2. Due to the lack of a suitable method to determine a true threshold value, this study introduced a reliable approach, which is called Hybrid method. The Hybrid method consists of two graphs, as following descriptions:

At the first stage, the method of mean residual life (MRL) plot (Fig. 2) is recommended to be evaluated for determining a true threshold value. The plot is a part of the GPD model for determining threshold value which was described in the previous section. In this study, for the first time, the MRL plot was employed for determining threshold value for the POT model. The prerequisite condition to employ the MRL plot is to secure the iid measurement for the data contributing in the analysis (Coles, 2001). The MRL plot is based on the determination of deviation from mean in the population. It is obvious that the extreme value analysis methods, such as the POT model, deal with standard deviation (errors of residuals) from the mean. Accordingly, the MRL plot can be employed in the POT analysis as the first part of Hybrid method to determine a true threshold value.

Second, the second part of the Hybrid method is done after determining the best fitting distribution function. Once the best fitting distribution function has been selected, several graphs of fitting for different threshold values should be drawn. For instance, in this study, several fitted Weibull distributions with shape parameter k = 1.4 have been depicted for different threshold values (Fig. 12). Ultimately, the most suitable threshold candidate(s) obtained from the first part (the MRL plot), should be assessed and evaluated. The thresholds, which were used in the figure can be chosen by considering the order and type of data. It is important to select the threshold values with considering the candidate(s) obtained from the MRL plot. Usually, 5 to 7 threshold values are sufficient to guide the user to make a suitable decision.

In most cases, the results of the first part of the Hybrid method determines a true threshold, however, the second part can be used for removing any misinterpretation of the MRL plot (more details in step (10)).

- 3. During evaluating a sample of extreme data, sometimes, an individual data exhibits a value much greater than other data. In the process of fitting the sample of data to a candidate distribution, that particular data is plotted at a position far above the line of the fitted distribution of return period curve. Such data is known as outlier(s). Therefore, the sample of extreme data contributing in the analysis should be examined to explore for the presence of outlier(s). The process of searching for outlier(s) can be done by several methods such as a statistical examination by Bamnett & Lewis (1994) or using of DOL (Deviation of OutLier) criterion method Goda (2000); Goda & Kobune (1993).
- 4. Calculating the two following essential sample parameters:
  - (a) Mean Rate  $\lambda$ , this parameter determines the number of observed data per year,  $\lambda = N_T/k$ . Where  $N_T$ , is the number of events and k, is the period of observed wave heights, in year.
    - (b) Censoring  $\nu$ , it determines a fraction of data that contributes in the analysis,  $\nu = N/N_T$ , in which N is the number of data applied in the analysis and  $N_T$ , number of the total data.
- 5. In this case, the POT model is applied with nine distribution candidates including FT-I, four FT-II distributions with fixed shape parameters 2.5, 3.33, 5, 10, and four Weibull distributions with fixed shape parameters 0.75, 1, 1.4, 2. Then, the

10

20

25

best fitting distribution for the data is selected among the nine distribution functions. Eqs. (8, 9, 10) represent the parent cumulative distribution functions (CDF),

(a) Gumbel Distribution (FT-I):  $-\infty