# Peer review of "Evaluation of Peaks-Over-Threshold Method"

_Ocean Science, 2016_

## Referee Comment (RC1) · Anonymous Referee #1 · 2 Aug 2016

The manuscript *Evaluation of Peaks-over-Threshold Method*, by Soheil Saeed Far and Ahmad Khairi Abd. Wahab, describes two extreme wave analysis models, namely Peaks-over-Threshold (POT) and Generalized Parete Distribution (GPD).

I am sorry that I have to recommend the editor the ***rejection*** of this manuscript, for the following reasons:

- The common nomenclature in extreme value theory is that there are two approaches: block maxima, or peak over thresholds. The block maxima are fitted to the GEV (Generalized EXtreme value) model, the peak over threshold values are fitted to the GPD (Generalized Pareto Distribution) model. It seems that this manuscript mixes up these words, and describes two models as POT and GPD, which - in my opinion - belong to each other: In the POT approach, the GPD is

fitted.

- The choice for one of those two models is completely based on a single dataset. The option that another dataset could possibly result in another conclusion, is not discussed.

- The Gumbel (FT-1) model is applied to peak-over-threshold values, which is wrong. The same holds for the FT-2 and Weibull models, which should be applied to block maxima.

- the manuscript lacks a good description of the wave data. There seem to be too many observations around 2.25m, 3.25 and 4.25 meter (see e.g. Figure 8). Especially the many observations around 4.25 m are suspect: one wouldn't expect so many values just below the maximum value. And if it were true, this points to a (physical) upper limit of the maxima.

- the choice of the Weibull shape parameter to be either 0.75, 1, 1.4 or 2 is rather arbitrary. The same holds for the FT-II distribution (fixed to 2.5, 3.33,5 or 10). The fixation of this shape parameter strongly influence the goodness-of-fit, and it also reduces the uncertainty range considerably. It would have been much more logical that the GEV (or GPD) distribution would have been fitted, in which also the shape parameters is estimated from the dataset, and its uncertainty influences the confidence bands. This is correctly done in section 3.1, but I don't understand what section 3.2 (the so-called POT method) adds to section 3.1.

---

## Author Comment (AC1) · 5 Aug 2016

Senior Scientist Andreas Sterl Topic Editor, Email: andreas.sterl@knmi.nl

Dear Dr. Sterl, I am pleased to inform you that we provided our point-by-point responses to the comments raised by the reviewer #1 dated August 2, 2016 for the following manuscript:

Journal: OS Title: Evaluation of Peaks-Over-Threshold Method Author(s): S. Saeed Far and A. K. Abd. Wahab MS No.: os-2016-47 MS Type: Research article

C: Comment & R: Response

1). C: The common nomenclature in extreme value theory is that there are two approaches: block maxima, or peak over thresholds. The block maxima are fitted to the

GEV (Generalized EXtreme value) model, the peak over threshold values are fitted to the GPD (Generalized Pareto Distribution) model. It seems that this manuscript mixes up these words, and describes two models as POT and GPD, which - in my opinion - belong to each other: In the POT approach, the GPD is fitted.

R: This manuscript is a part of an academic research, which mainly aims to improve the Peaks-Over-Threshold (POT) model, introduced by Goda (1988). The POT model is completely different with the GEV or GPD models described by Coles (2001). Despite of some accusations have been seen about the POT model, however, it has commonly been used since the last two decades. In addition, the results of the POT model have shown its credibility compared with the modern models such as the GPD model. The start point of this research turns back to 2012, it happened after publishing a paper by Li et al. (2012). Li and his co-authors compared three extreme wave models (GEV, POT and GPD) and concluded that POT and GPD provide credible results. However, the authors, despite of their results, left many other questions behind without providing suitable answers. This manuscript is an effort to answer some of the important questions concerning to the POT and GPD models, and introducing two proposed methods to improve the POT model.

2). C: The choice for one of those two models is completely based on a single dataset. The option that another dataset could possibly result in another conclusion, is not discussed.

R: Despite of obtaining several results, this manuscript described a comparison between POT and GPD. The aim of comparing the two models was to evaluate the models in order to highlight their merits and limitations. Therefore, we needed to employ same data in developing the two models. This provides a suitable platform to make a comparative study like the published research by Li at el. (2012).

3). C: The Gumbel (FT-1) model is applied to peak-over-threshold values, which is wrong. The same holds for the FT-2 and Weibull models, which should be applied to

block maxima.

R: Please see Goda (1988); Goda (2000) and Goda (2010). This is not the GEV model.

4). C: the manuscript lacks a good description of the wave data. There seem to be too many observations around 2.25m, 3.25 and 4.25 meter (see e.g. Figure 8). Especially the many observations around 4.25 m are suspect: one wouldn't expect so many values just below the maximum value. And if it were true, this points to a (physical) upper limit of the maxima.

R: This comment is not completely clear. The majority of the observed data are in the range of 2 to 3.5 m. However, around 4.25 m less data were recorded. In the manuscript, we did not use the method of Block Maxima, and the GEV model has not been employed or noted.

5). C: the choice of the Weibull shape parameter to be either 0.75, 1, 1.4 or 2 is rather arbitrary. The same holds for the FT-II distribution (fixed to 2.5, 3.33,5 or 10). The fixation of this shape parameter strongly influence the goodness-of-fit, and it also reduces the uncertainty range considerably. It would have been much more logical that the GEV (or GPD) distribution would have been fitted, in which also the shape parameters is estimated from the dataset, and its uncertainty influences the confidence bands. This is correctly done in section 3.1, but I don't understand what section 3.2 (the so-called POT method) adds to section 3.1.

R: I think, this comment has been based on the assumption of using the GEV model. The use of fixed shape parameters has never been the manuscript's idea. It is a routine procedure of developing the POT model. We know that all of these statistical models come with uncertainties, and sometimes the criticisms are reasonable, however, a perfect model with no uncertainty has not been introduced, yet.

We appreciate your time taken to review the responses.

Yours Sincerely, Soheil Saeed Far
**OSD**
References:

1. Goda, Y. (2010). Random seas and design of maritime structures. World scientific.
2. Goda, Y. (2000). Random seas and design of maritime structures. World scientific.
3. Goda, Y. (1988). On the methodology of selecting design wave height. Coastal Engineering Proceedings, 1(21). 4. Coles S. (2001). An introduction to statistical modeling of extreme values. London: Springer Series in Statistics. 5. Li, F., Bicknell, C., Lowry, R., & Li, Y. (2012). A comparison of extreme wave analysis methods with 1994-2010 offshore Perth dataset. Coastal Engineering, 69, 1-11.

Please also note the supplement to this comment:
http://www.ocean-sci-discuss.net/os-2016-47/os-2016-47-AC1-supplement.pdf

———————————————————

---

## Referee Comment (RC2) · Anonymous Referee #2 · 15 Aug 2016

General evaluation

Although the manuscript addresses an interesting topic, it has some problems and as such I do not recommend its publication.

Comments

1) The authors claim that the GPD and the POT are two different models. The GPD plays a crucial role in extreme value theory as the distribution of the sample of excesses above a sufficiently high threshold, method known as the peaks-over-threshold (POT) - see, for instance, Davison and Smith (1990), Pickands (1975), Embrechts et al (1997). The POT and the GPD are thus closely connected.

I do not understand the meaning of "to select the best fitting distribution for the dataset"
(page 1, line 23). The issue is to fit a GPD distribution to the excess (or exceedance) data. Obviously, the Generalized Extreme Value (GEV) or the Gumbel, Fréchet and Weibull are candidate distributions if the Annual Maxima Method is considered. For using this method the data has to exhibit a temporal structure (and obviously there is no threshold involved).

(Some references of using the POT in waves context - Teena et al (2012), Thevasiyani and Perera (2014) and Cañellas (2007))

2) The authors, in page 6, present the hybrid method that starts with the choice of the threshold through the mean excess (ME) plot. In applications, the choice of the threshold just by looking at the ME plot is generally very difficult if not impossible (in some cases). Due to the practical difficulties in choosing the threshold, and considering the importance of this step in the method, some complementary approaches could have been mentioned (see, for instance, Beirlant et al (2004)). In page 7-line 11, the authors say that Weibull distributions were fitted to the wave data. Later on, the same is done for the Gumbel and for the Fréchet distributions. Due to the fact that "... the exceeded data over a certain threshold are employed in the analysis" (page 9, line 10) then the GPD should have been used instead because it is the proper distribution for modelling the excesses above the threshold, as was stated in my previous comment.

3) The data lacks a convenient explanation. The authors seem to consider wave heights recorded at time intervals of at least three days apart (page 2, lines 28-29). Then, the authors state that they do linear interpolation "to fill in the large gaps" (line 29, page 2). I wonder why that is needed. The exceedances are supposed to be the focus. Additionally, it would have been nice to see the plot of the data which was analysed.

Technical correction

Expression (4) is not consistent with the parameterization of the GPD the authors indicated in (1) – see Coles (2001).

**References**

Beirlant J, Goegebeur Y, Segers J, Teugels J (2004). Statistics of Extremes – Theory and Applications. Wyley, Chichester.

Davison AC and Smith RL (1990). Models for exceedances over high thresholds. J Roy Stat Soc B 52:393-442.

Embrechts P, Klüppelberg C, Mikosch, T (1997). Modelling Extremal Events for Insurance and Finance. Springer-Verlag Berlin Heidelberg.

Pickands III, J (1975). Statistical inference using extreme order statistics. Ann Stat 3:119-131. Teena, NV, Kumar S, Sudheesh K, Sajeev R (2012) Statistical analysis on extreme wave height. Natural Hazards, Volume 64, Issue 1, pp 223-236.

Thevasiyani T, Perera, K (2014) Statistical analysis of extreme ocean waves in Galle, Sri Lanka. Weather and Climate Extremes, Volumes 5–6, October 2014, Pages 40–47.

Cañellas B, Orfila A, Méndez FJ, Menéndez M, and Tintoré J (2007) Application of a POT model to estimate the extreme significant wave height levels around the Balearic Sea (Western Mediterranean). Journal of Coastal Research, Special Issue 50.

---

## Author Comment (AC2) · 21 Aug 2016

**Soheil Saeed Far**
*Universiti Teknologi Malaysia*
✆ *+60 (17) 826 8048*
✉ *soheilsaeedfar@gmail.com*

**Senior Scientist Andreas Sterl**                                    August 21, 2016
*Topic Editor,*
*Email: andreas.sterl@knmi.nl*

Dear Dr. Sterl,

I am writing to inform you that we provided our point-by-point responses to the comments raised by the reviewer #2 dated August 15, 2016 for the following manuscript:

Journal: OS
Title: Evaluation of Peaks-Over-Threshold Method
Author(s): S. Saeed Far and A. K. Abd. Wahab
MS No.: os-2016-47
MS Type: Research article

**C:** Comment & **R:** Response

1. **C: 1.1. The authors claim that the GPD and the POT are two different models.**

   R: We do not claim that the GPD and POT (Goda) are different models. They are different models. The POT model was introduced by Goda (1988) and then was developed in a textbook (Goda, 2000). The new edition of the book was published in 2010 (Goda, 2010). This author was referred $23^{rd}$ times throughout our manuscript and the first 7 references in the list of references have been allocated to the Goda's publications. For example in page 6, line 17, title 2.2, Peaks-Over-Threshold Model introduced as Goda method (this is where the methodology of POT was described), or in the line of keywords (Pg. 1, line 12), the POT model accompanied by the name of Goda in parentheses. Therefore, the manuscript has left no any gap of explanation or any excuse to misunderstanding between the POT (Goda) model and other models.

   **C: 1.2. The GPD plays a crucial role in extreme value theory as the distribution of the sample of excesses above a sufficiently high threshold, method known as the peaks-over-threshold (POT) -see, for instance, Davison and Smith (1990), Pickands (1975), Embrechts et al (1997).**

   R: The publications of Pickands, Davison and Smith have been cited in our manuscript, and we used their ideas. We were aware about the similarity of the names (POT), that is

why many times the name of Goda were noted beside the POT model.

**C: 1.3. The POT and the GPD are thus closely connected.**

R: The POT (Goda) and GPD models use excesses data over a sufficiently high threshold value. GPD is an asymptotic model introduced by Picknads (1975), apart from this, the POT model for the first time introduced by Goda (1988). [All these information have been discussed and noted throughout the manuscript.]

**C: 1.4. I do not understand the meaning of "to select the best fitting distribution for the dataset" (page 1, line 23).**

Based on definitions, in the extreme data analysis, many theoretical distribution functions are employed for fitting to samples. In theoretical statistics, a data of extremes refers to the maximum or minimum among a sample of independent data. When extreme analysis is applied for a sample of such extreme data, it is known that three types of theoretical functions should fit such samples, depending on the population distribution of initial data. However, the data of extreme wave heights collected by POT are different from the extreme data of theoretical statistics (more details, Goda [Pg. 377](2000) Chapter 11, Statistical Analysis of Extreme Waves).

**C: 1.5. The issue is to fit a GPD distribution to the excess (or exceedance) data. Obviously, the Generalized Extreme Value (GEV) or the Gumbel, Fréchet and Weibull are candidate distributions if the Annual Maxima Method is considered. For using this method the data has to exhibit a temporal structure (and obviously there is no threshold involved).**

R: In the manuscript, we have not employed the GEV model. The reason of using the three **FT-I**, **FT-II** and **Weibull** distribution functions is to conduct the POT model, and figure out the best fitting distribution function for the dataset.

**C: 1.6. (Some references of using the POT in waves context - Teena et al (2012), Thevasiyani and Perera (2014) and Cañellas (2007))**

R: There are also many references of using the POT (Goda) model such as Li et al. (2012), Goda (2010) and Goda (2004).

**C: 2.1. The authors, in page 6, present the hybrid method that starts with the choice of the threshold through the mean excess (ME) plot. In applications, the choice of the threshold just by looking at the ME plot is generally very difficult if not impossible (in some cases). Due to the practical difficulties in choosing the threshold, and considering the importance of this step in the method, some complementary approaches could have been mentioned (see, for instance, Beirlant et al (2004)).**

R: In the manuscript, the Hybrid method was proposed to determine true threshold value for the POT model. The method consists of two parts: the first part is using the Mean

Residual Life (MRL) plot, and the second part, which is described as the complementary method can be found in page 7 line 9 to 17, and step 10 in page 10. Therefore, the Hybrid method consists of **exploratory** and **complementary** methods.

The discussion of credibility of extracting true results via the MRL plot can be found in several literatures for instance Coles (2001) discussed about the interpretation of the graph, and noted that, "The interpretation of a mean residual life plot is not always simple in practice." To my knowledge, the use of MRL and extracting true result from the plot is intuitively understandable if the definition of deviation from mean in the population has already been understood.

**C: 2.2. In page 7-line 11, the authors say that Weibull distributions were fitted to the wave data. Later on, the same is done for the Gumbel and for the Fréchet distributions. Due to the fact that ". . . the exceeded data over a certain threshold are employed in the analysis" (page 9, line 10) then the GPD should have been used instead because it is the proper distribution for modelling the excesses above the threshold, as was stated in my previous comment.**

R: That is the explanation of step 8, which describes using the excess data to determine the plotting position for the POT model. The methodology of POT initiates from page 6, section 2.2 under the title of, "Peaks-Over-Threshold Model (Goda Method)" to page 11, section 3, entitled, "Results and Discussions".

**C: The data lacks a convenient explanation. The authors seem to consider wave heights recorded at time intervals of at least three days apart (page 2, lines 28-29). Then, the authors state that they do linear interpolation "to fill in the large gaps" (line 29, page 2). I wonder why that is needed. The exceedances are supposed to be the focus. Additionally, it would have been nice to see the plot of the data which was analysed.**

R: The process of choosing data in the time intervals of three days were done on the data to secure the required independence of the data. It has been done after data collection and prior to the determination of threshold value. After the process of data collection, usually some gaps of missing data for several days or more happen due to human error or instrument malfunction. Those gaps were filled by linear interpolation. This process is done before exerting the time intervals (three days) on the data.

Based on the regulation of using the data, we are not allowed to distribute or publish the data. However, the complete data analysis have been done, and this is not the first evaluation on this data. If any specific graph is required, which its distribution does not violate the regulations, please let me know, then I will gladly provide the arrangements to get permission to send it for you.

2. **Technical correction**
   **C: 3. Expression (4) is not consistent with the parameterization of the GPD**

**the authors indicated in (1) – see Coles (2001).**

R: Yes, it happened during the process of typesetting. The correct expression is,

$$x_m = u - \frac{\sigma}{\xi} \left[ (m \cdot \zeta_u)^{-\xi} - 1 \right] \tag{4}$$

And, then equation 7,

$$x_m = u - \frac{\sigma}{\xi} \left[ (ARI \cdot \frac{N}{k})^{-\xi} - 1 \right] \quad for \ (\xi \neq 0) \tag{7}$$

We appreciate your time taken to review the responses. Lot of thanks for your consideration and care.

Yours Sincerely,

**Soheil Saeed Far**

Goda, Y. (2010). Random seas and design of maritime structures. World scientific.

Goda, Y. (2000). Random seas and design of maritime structures. World scientific.

Goda, Y. (1988). On the methodology of selecting design wave height. Coastal Engineering Proceedings, 1(21).

Coles S. (2001). An introduction to statistical modeling of extreme values. London: Springer Series in Statistics;

Li, F., Bicknell, C., Lowry, R., & Li, Y. (2012). A comparison of extreme wave analysis methods with 1994-2010 offshore Perth dataset. Coastal Engineering, 69, 1-11.

Pickands J. (1975). Statistical inference using extreme order statistics. the Annals of Statistics. 1:119-31.

Goda, Y. (2004). Spread parameter of extreme wave height distribution for performance-based design of maritime structures. Journal of waterway, port, coastal, and ocean engineering, 130(1), 29-38. URL: http://ascelibrary.org/doi/abs/10.1061/(ASCE)0733-950X(2004)130:1(29)

---

## Editor Comment (EC1) · A. Sterl (Editor) · 31 Aug 2016

Dear dr Far, dear dr. Wahab,

The discussion period of your paper "Evaluation of Peaks-Over-Threshold Method" is over. Only the two anonymous reviewers have submitted a comment, and you have answered them.

The points raised by the reviewers are very serious, and to my judgement your answers fail to fully address their concerns. The main problems are

- The distinction between GPD and POT (Goda) is not clear. Both rely on fitting the exceedences over a threshold to a theoretical distribution function.

- Using a GPD to do so (which is usually called POT method in the literature) is

based on solid mathematical theory. I do not see the bases for a fitting to the FT-I/II/III distributions. As both reviewers point out the FT-I/II/III (or GEV) distributions are the theoretical distributions for block-maxima and not for exceedences.

- It is unclear to me why you use fixed values for the shape parameter (see also reviewer #1) instead of estimating them as part of the fitting procedure.

- Both reviewers point at problems with your data. Reviewer #1 points out that there are suspiciously many data points close to 4.4 m (Fig. 8), but your answer fails to adequately explain this occurrence. Your answer that the GEV method has not been employed is off the point.

- In one of your answers to reviewer #1 you state that you are not allowed to show your data. This is unacceptable. Science relies on the reproducibility of results, and one cannot reproduce your results without having the original data, nor can one judge on their quality. As an Open Access journal, Ocean Science is dedicated to the openness of science and the scientific process. Publishing results based on disclosed data contradicts this idea.

Based on these considerations I regret that I cannot encourage you to submit a revised version of the paper.

With kind regards,
Andreas Sterl

---

## Short Comment (SC1) · 1 Sep 2016

Dear Dr. Sterl,

Thanks for your letter dated, Aug 31, 2016.

I regret to inform you that I do not agree with the two reviewers' comments on which your decision was based due to the following reasons:

First, I declare that the reason of writing this letter is to respond the comments and misunderstanding about our manuscript, and we respect your decision as the Topic Editor of the journal Ocean Science.

There are some key points that help to understand our study. However, I am wonder why they are ignored. For instance in the methodology section, we separated the two

models into two subsections and both models have their separate references (Goda, 2010) and (Coles, 2001). However, the reviewers continuously mixed the models with each other.

The use of FT-I (Gumbel), FT-II (Frechet) and Weibull distributions and using fixed shape parameter are the process of Goda model (Please see our second reply). We improved the POT (Goda) model. I have not yet seen any question about the manuscript's objectives or conclusions! It is not very difficult to google the method and figure out something about the POT (Goda) model.

About the data:

I think, we should first settle in the matter of existing the POT (Goda) model, and then we can talk about the data. This method exists with or without the reviewers' acknowledgement.

In your letter, you mentioned that, 'Reviewer #1 points out that there are suspiciously many data points close to 4.4 m (Fig.8), but your answer fails to adequately explain this occurrence. Your answer that the GEV method has not been employed is off the point'.

The reviewer's comment – regard to his/her previous words – has another question in the connotation of his/her comment. I am sure the reviewer #1 has got his/her answer from my response; the reviewer #1 wanted to know about the group of data close to the highest data (4.25 m) in the sample of statistics, which are considered in the block-maxima used by the GEV model. My response was an attempt to terminate the assumption of using GEV in the manuscript. Nevertheless, in the reply letter, I rejected the use of many data around 4.25 m (the response in reply 1: The majority of the observed data are in the range of 2 to 3.5 m. However, around 4.25 m less data were recorded.).

Ultimately, you quoted one of my responses (in my second reply) about the unavailability of data for publication and distribution, and then you explained that the whole work

is under the question if the data were not available.

Dear Editor, you are an experienced and academic person, who dedicated many years of his life in the academic atmosphere. Regard to this background, during the study of your letter, first, I thought you were probably very busy, so you trusted the reviewers' opinions. However, terminating your letter with discussion about the availability of data, and ignoring all our reasonable responses, as well as ignoring the gap of reviewers' knowledge about the POT (Goda) model, and then dragging the manuscript down to a low level and beating it with experience by talking about an ambiguous question about the suspicious data (!) was not my expectation from an academic person.

You are well aware that there are several ways to distribute such dataset similar to our case. However, when there is no any sign of understanding the study, it would not be a reasonable decision to manage all the eggs in one basket.

Thank you very much for your time and consideration.

Best Regards,

Soheil Saeed Far